# Fabrication of Microfluidic Chips Based on an EHD-Assisted Direct Printing Method

**DOI:** 10.3390/s20061559

**Published:** 2020-03-11

**Authors:** Xiang Chi, Xinyu Zhang, Zongan Li, Zhe Yuan, Liya Zhu, Feng Zhang, Jiquan Yang

**Affiliations:** Jiangsu Key Laboratory of 3D Printing Equipment and Manufacturing, School of NARI Electric and Automation, Nanjing Normal University, Nanjing 210046, China; Xchi111@163.com (X.C.); zxy334010@163.com (X.Z.); yuanzhe199504@163.com (Z.Y.); 61193@njnu.edu.cn (L.Z.); fengzhan@njnu.edu.cn (F.Z.); jiquany@126.com (J.Y.)

**Keywords:** EHD-assisted direct printing, microfluidic chip, paraffin wax mold, micromixing

## Abstract

Microfluidic chips have been widely used in many areas such as biology, environmental monitoring, and micromixing. With the increasing popularity and complexity of microfluidic systems, rapid and convenient approaches for fabricating microfluidic chips are necessary. In this study, a method based on EHD (electrohydrodynamic)-assisted direct printing is proposed. Firstly, the principle of EHD-assisted direct printing was analyzed. The influence of the operating voltage and moving speed of the work table on the width of a paraffin wax model was studied. Then, two kinds of paraffin wax molds for micromixing with channel widths of 120 μm were prepared. A polydimethylsiloxane (PDMS) micromixer was fabricated by replicating the paraffin wax mold, and the micromixing of blue and yellow dye was realized. The results show that EHD-assisted direct printing can be used to make complex microscale structures, which has the potential to greatly simplify the manufacturing process.

## 1. Introduction

Microfluidic chips were developed as gas chromatographs in the 1970s. They have the main feature of manipulating fluids in micro- or nanoscale space and have been used for micromixing and separation purposes [1,2,3]. Polydimethylsiloxane (PDMS), a silicon-based polymer, has been widely used in miniaturized total analysis systems (μ-TAS) for the fabrication of microfluidic devices as it has the advantage of low cost. PDMS can also be bonded to itself and other substrates easily; it is optically transparent, durable, non-fluorescent, biocompatible, chemically inert, and nontoxic; and sealing PDMS is simpler than sealing glass or silicon. Therefore, PDMS is often the preferred material for prototyping and testing various microfluidic devices [4,5].

The most commonly used technique for fabricating PDMS devices is soft lithography, which is a non-photolithographic approach for micro- or nanofabrication based on replica molding [6,7,8,9]. To make the soft lithography method easy to use and suitable for large-scale production, the mold fabrication material should be low cost and easily available, and the process should be highly efficient. To meet the demand for personalization, the method requires high flexibility. Multiple steps for the fabrication of molds, such as copper sheet etching [10], liquid molding [11], gelatin casting [12], two-stage embossing [13], Shrinky-Dink molding [14], and screen printing [15], are required to make the template at the first step. The nylon thread method [16] has high resolution and is quite simple, but ink jet printing [17] and 3D-printed molds [18] can be fabricated directly from a computer-aided design (CAD) pattern.

Fused deposition modeling (FDM) is based on the extrusion of polylactic acid (PLA) and acrylonitrile–butadiene–styrene copolymer (ABS)-based material out of a nozzle with a diameter ranging from 50 to 400 μm [19], and sugar or liquid metal is also used as the extruding material. Its outstanding features are the low cost of equipment, wide popularity, and simple operation. Parekh et al. [20] fabricated 3D microfluidic channels using printed liquid metal on a PDMS sheet; the liquid metal in the cured PDMS was removed by an electrochemical reaction, leaving the 3D microchannels. He et al. [21] printed melting sugar using a modified FDM printer; assisted by PDMS solution printing, the sugar could be sacrificed. A microchannel with a semi-circular cross section was realized. Extrusion-based printing methods effectively simplify the machining process and significantly reduce the technical threshold and machining cost, but the resolution still needs to be improved and the mold and support material are difficult to remove.

Another method to deposit a mold according to a computer-aided design (CAD) pattern is based on the electric field. Electrohydrodynamic (EHD) printing uses the electrostatic force on the nozzle tip to form a Taylor cone; in the Pyro-EHD method, the pyroelectric field of a lithium tantalate or lithium niobate crystal is activated by a wireless thermal source, and the printing material drop starts to deform into a Taylor cone under the action of the pyroelectric charges [22,23,24]. Both these methods have the feature of high manufacturing accuracy, especially the Pyro-EHD method, which does not require electrodes, high-voltage circuit connections, or nozzles. Han et al. [25] realized wax microdroplets with a size of 8 μm via EHD printing, and Lan [26] modified the EHD printing method to suit multiple scales and multiple materials. Coppola et al. [27] produced a Pyro-EHD printed poly (lacticco-glycolic acid, 3200 cps) PLGA male mold and prepared single-layer grid PDMS channels.

In this paper, an EHD-assisted direct printing method for the sacrifice molding of microfluidic chips is studied. Paraffin wax was printed on the substrate and replicated to make a PDMS microfluidic chip. The EHD-assisted direct printing system, especially the heating devices for the nozzle and the principle of the EHD-assisted direct printing, was studied. Then, the influence of the voltage and travelling velocity of the printing head on the paraffin wax mold was researched, a paraffin wax mold was printed with the optimized parameters, and a kind of microfluidic chip for micro mixing was fabricated. Finally, the mixing of blue dye and yellow dye solution was realized on the fabricated PDMS microfluidic chip. The paraffin wax used is low cost, easily available, and easy to sacrifice. The EHD-assisted directing printing method has the high efficiency of extrusion together with the high accuracy of EHD printing.

## 2. Materials and Methods

### 2.1. Experimental Materials

A model 34G metal conductive nozzle with an inner diameter of 60 μm, outer diameter of 250 μm, total length of 25 mm, and tube length of 13 mm was obtained from the Aroh Alona Company. The paraffin wax was provided by Shenzhen Hengsheng Biotechnology Company, with properties shown in Table 1.

### 2.2. Experimental System

The EHD-assisted direct printing system includes an AFG 2021C signal generator from Taike co., Ltd. and high-voltage DMC-200 power source (0–4 KV) from Dalian Dingtong Company, a high-precision xyz platform, a paraffin-wax-containing part with a syringe, and a 34G metal nozzle from the Aroh Alona Company, and a printing assistant part with a heating device and a pressure control device, as shown in Figure 1.

### 2.3. The EHD-Assisted Direct Printing Process

The paraffin wax was melted, filtered through a 1000 mesh screen, and fed into the syringe. Then, the syringe and the nozzle part were heated to 90 °C and 150 °C separately and kept at that temperature. To make the paraffin wax flow out of the nozzle continuously, air pressure was applied to the wax container. Affected by both the air pressure and gravity, a meniscus shape formed in the nozzle tip. When the voltage was applied between the nozzle and the substrate, an electric field was formed between the nozzle and the motion stage. The meniscus formed a Taylor cone with increasing voltage, and when the electrostatic force exceeded the surface tension, the paraffin wax was printed from the bottom of the Taylor cone as shown in Figure 2.

### 2.4. The Microfluidic Chip Fabricating Process

The microfluidic channel was modeled using CAD software and converted into a file format suitable for processing by the EHD-assisted direct printing system. Using this method, various microfluidic channel structures can be designed easily and quickly. The glass substrate was cleaned with concentrated sulfuric acid, acetone, and deionized water, and the paraffin wax model was printed on the glass substrate. Liquid silicone elastomer and a special curing agent for the PDMS elastomer with a volume ratio of 10:1 were thoroughly mixed with a magnetic stirrer at a speed of 120 rpm and then degassed in a vacuum drying oven. The PDMS liquid was slowly poured onto the paraffin wax channel model, and the glass substrate was placed in the drying oven at 40 °C for 24 h to cure the PDMS. The cured PDMS chip was then cut along the boundary and stripped. The PDMS chip was immersed in a cyclohexane solution for 10 min, washed in deionized water, and then dried with nitrogen gas. After that, the PDMS chip was immersed in 80 °C deionized water for 10 min. The chip was then taken out quickly, washed in an ethanol analytical reagent (AR), cleaned by nitrogen gas, and dried in an oven at 85 °C for 20 min. Then, the edges of the chip were cut carefully, the burrs were removed, and PDMS chips were drilled and cleaned. Finally, the PDMS devices were plasma treated and bonded with another clean piece of glass by pressing and maintaining at 75 °C for 30 min. The whole process is schematically shown in Figure 3.

## 3. Results

### 3.1. Dynamic Changes of Droplets at the Nozzle

A universal serial bus (USB) electron microscope was used to observe the real-time process of EHD-assisted direct printing. Figure 4 shows the dynamic change in the droplet shape at the nozzle tip when paraffin wax was being printed. As paraffin wax is a phase change material, the parameters were adjusted according to the melting state of the paraffin wax. When air pressure was applied, affected by both the air pressure and gravity, a meniscus shape formed in the nozzle tip, as shown in Figure 4a. Precise temperature control equipment was used. The temperature at the nozzle tip was set to 150 °C, and the molten paraffin wax had high fluidity at this temperature. The air pressure parameters were adjusted according to the actual situation by using a pressure regulating valve.

Since the nozzle was connected with the positive pole of the high-voltage power supply and the motion stage was connected with the negative pole, an electric field was formed between the nozzle and the motion stage. The shape of the droplet was stretched from a meniscus shape under the action of the electric field force and formed a Taylor cone, as shown in Figure 4b. This kind of EHD-assisted direct printing method utilizes the phenomenon between the Taylor cone and the total extrusion printing.

### 3.2. Analysis of Experimental Parameters

#### 3.2.1. Working Voltage

The working voltage is one of the key factors affecting the print resolution. The printing parameters were set as follows. A metal needle with a 60 μm inner diameter and 250 μm outer diameter was used. The temperature was set to 90 °C at the syringe and 150 °C at the nozzle. The nozzle tip was 500 μm from the substrate, the printing speed was 10 mm·s^−1^, and the working voltage was 1000–2000 V. The EHD-assisted direct-printed paraffin wax lines obtained under the conditions of increasing working voltages from 1000 to 2000 V are shown in Figure 5.

When the voltage was too low, the electric field force of the micromolten wax was less than the surface tension, and jetting could not be formed. When the voltage reached 1000 V, cone printing could be formed, but the electric field force of the nozzle tip was insufficient. Due to the phase transformation phenomenon, the paraffin wax solidifies quickly once it contacts with the glass substrate. The printing process using the tail of the Taylor cone is not stable enough for continuous working. When the voltage range was from 1100 to 1800 V, stable cone printing could be formed for a reliable wax line by utilizing the middle part of the Taylor cone. However, when the voltage exceeded 1900 V, the cone printing was affected by too large an electric field and the printing process was unstable, causing dispersed printing and making it difficult to achieve accurate and controllable printing. The relationship between the line width and working voltage is shown in Figure 6. In summary, when the voltage was 1000 V, the printing process was unstable. Continuous and stable printing could be achieved between 1100 and 1800 V, but the line width was slightly different. When the voltage was above 1900 V, the printed lines were nonuniform. The working voltage had a significant effect on the print shape, which in turn affected the print resolution and line width. The working voltage range applicable for stable printing is wide, which means that the quality and continuity of printed microfilaments were almost not affected by the change of the voltage within the feasible range.

#### 3.2.2. Printing Speed

For the continuous print mode, the formation of the paraffin wax lines was different at different moving speeds, and the width of the paraffin wax line was also obviously influenced. We set the material supply device temperature to 90 °C, the needle temperature to 150 °C, the distance between the nozzle tip and the substrate to 500 μm, the working voltage to 1300 V, and the printing speed to 1, 5, 10, 20, 30, 40, or 50 mm·s^−1^. The printing results are shown in Figure 7, the line width was measured using a high-precision optical microscope, and the relationship between the line width and printing speed is shown in Figure 8.

After the paraffin wax was printed on the glass substrate, due to a phase transformation of the paraffin wax and its adhesion to the substrate, viscous drag force was generated in the printing process. When the printing speed was too low, the pulling force caused by the viscous drag force was small, and the influence on the line width was small. As the printing speed increased, the pulling force caused by the viscous drag force gradually increased and the line width was correspondingly reduced. When the printing speed was 50 mm/s, the printing was unstable. In summary, increasing the printing speed will reduce the paraffin wax line width, and the paraffin wax line was continuous within a defined printing speed range.

### 3.3. PDMS Microchannels Replicated Using the Paraffin Wax Mold

The results showed that with optimized system parameters for EHD-assisted direct printing, a controllable microfilament size could be printed. The line width has an important influence on the resolution and performance of the printed structure. By controlling the process parameters, the patterns shown in Appendix A could be constructed. Microcrystalline paraffin wax was selected as the printing material (material properties are shown in Table 1). The Y-type and zigzag paraffin wax molds were prepared by adjusting the system parameters and replicated to prepare PDMS microchannels. The PDMS microchannels were observed by a scanning electron microscope (SEM) as shown in Figure 9. The intersection area of the PDMS microchannel had a greater height than the other areas due to overlapping of the repeatedly printed paraffin wax.

A SEM photo of the straight area of the PMDS microchannel replicated from the paraffin wax mold is shown in Figure 10. The surface of the microchannel was smooth. However, due to the rapid solidification of the molten paraffin wax contacting the cold glass substrate, the edge of the microchannel exhibited a zigzag pattern. The mixing of fluid usually depends on the generation of chaotic advection and/or turbulence, in which the fluid motion varies irregularly and thus causes quantities such as the pressure and velocity to vary randomly in both space and time. The simplest method of obtaining chaotic advection is to insert obstacles into the mixing channel. The zigzag structure could play the role of an obstacle to some extent and thus cause some imbalances of the flowing fluid and speed up the micromixing process [28,29].

The determined line width could be realized via the EHD-assisted direct printing method through defined system parameter groups, and this method can also be used to quickly and flexibly prepare paraffin wax patterns with different widths according to actual needs, which would greatly improve the application of this kind of method for microfluidic chips. Microchannels with different widths and heights were prepared by adjusting system parameters, as shown in Figure 11. The cross section of the microchannel was a semi-circular profile. The system parameters were set as follows. The nozzle tip was 500 μm from the substrate. In Figure 11a, the printing speed was 5 mm·s^−1^, and the working voltage was 1500 V. The width of the microchannel was 270 μm and the height was 105 μm. In Figure 11b, the printing speed was 1 mm·s^−1^, the working voltage was 1800 V, the microchannel width was 440 μm, and the height was 180 μm.

The diminutive scale of the flow channels in microfluidic systems increases the surface-area-to-volume ratio. Y-shaped and zigzag micromixers were designed and fabricated according to the process stated in the previous part using 120 μm paraffin wax molds. The micromixing experiment was performed on two kinds of PDMS microfluidic chips. Two channel syringe pumps were used to feed the fluid. Quantities of 800 μM erioglaucine (blue dye) and 1870 μM tartrazine (yellow dye) were prepared and injected into the inlet channels of the passive micromixer at a speed of 10 μL·min^−1^. In the Y-shaped and zigzag microchannel, the blue dye and yellow dye solutions showed a clear interface at the junction at a feed speed of 10 μL·min^−1^, as shown in Figure 12 and Figure 13. The area of the interface increased slowly as the fluid flowed; the color of the fluid near the outlet channel of the Y-shaped microchannel was a mixture of yellow and blue (Figure 12c), and the color of the fluid was green in the outlet channel of the zigzag microchannel (Figure 13c). It can be seen that the reasonable design of bending in the microchannel structure can make the fluid mix faster. This method is a fast and easy fabrication method for the preparation of wax molds used in the replication of PDMS microchannels.

## 4. Conclusions

In this paper we proposed an EHD-assisted direct paraffin wax printing method for microfluidic chips. This method utilizes both EHD 3D printing and heated paraffin wax extrusion. The paraffin wax was melted and extruded out of the nozzle with the assistance of an electric field force of up to 1800 V to form a paraffin wax mold on a glass substrate. The wax mold was replicated with PDMS and cured. A microchannel was drilled and bonded to the glass substrate to prepare a microfluidic chip. To avoid blocking and to ensure printing continuity, the syringe and the nozzle tip were heated to 90 °C and 150 °C, respectively. With a 60 μm metal nozzle, when the printing speed was 10 mm/s, the applicable working voltage ranged from 1100 to 1800 V. Paraffin wax lines with width ranging from 80 to 220 μm and height ranging from 35 to 90 μm can be prepared using this method. In this range, the width and height of the paraffin wax line increased with increasing working voltage. The Y-shaped and zigzag micromixers with channel widths of 120 μm were prepared and the micromixing of blue dye and yellow dye was realized. This method has the advantages that paraffin wax is easily available and low cost, electric field force extrusion has high controllability and efficiency, and the fabricated paraffin wax mold has high accuracy and is easy to sacrifice. A more precise wax mold may be prepared using nozzles with smaller diameter, and this method can be further used to fabricate microfluidic chips for many other applications.

## Figures and Tables

**Figure 1 sensors-20-01559-f001:**
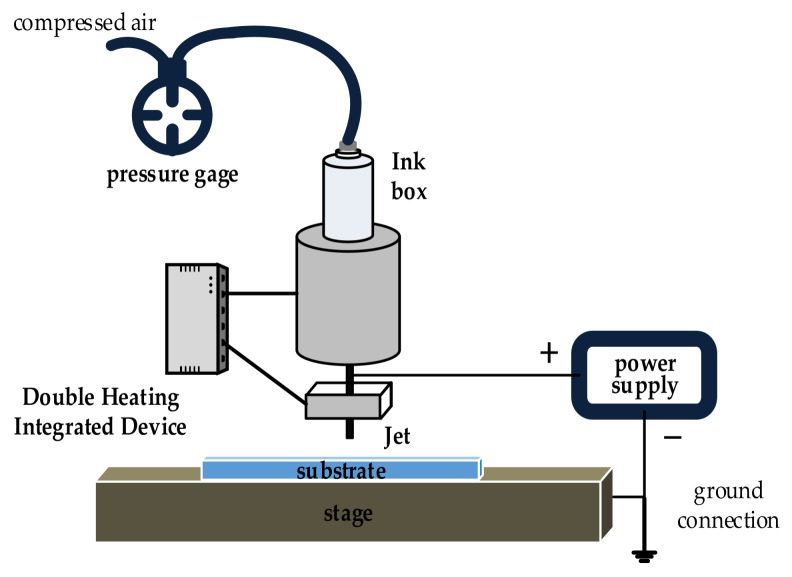
A schematic diagram of the electrohydrodynamic (EHD)-assisted direct printing system.

**Figure 2 sensors-20-01559-f002:**
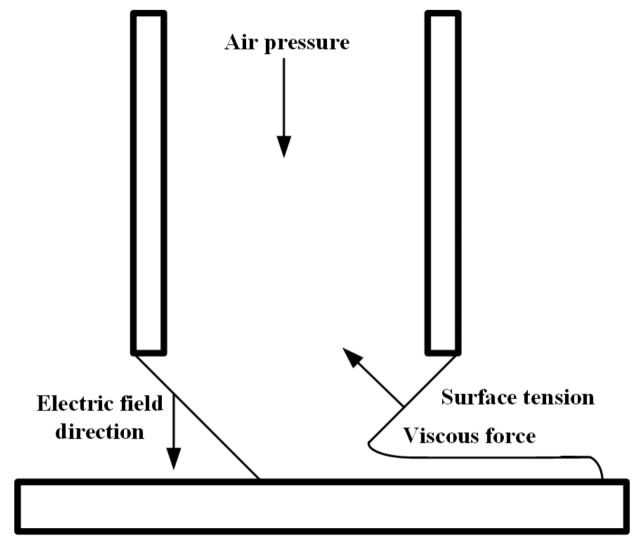
Schematic diagram of the basic principle of the EHD-assisted direct printing system.

**Figure 3 sensors-20-01559-f003:**
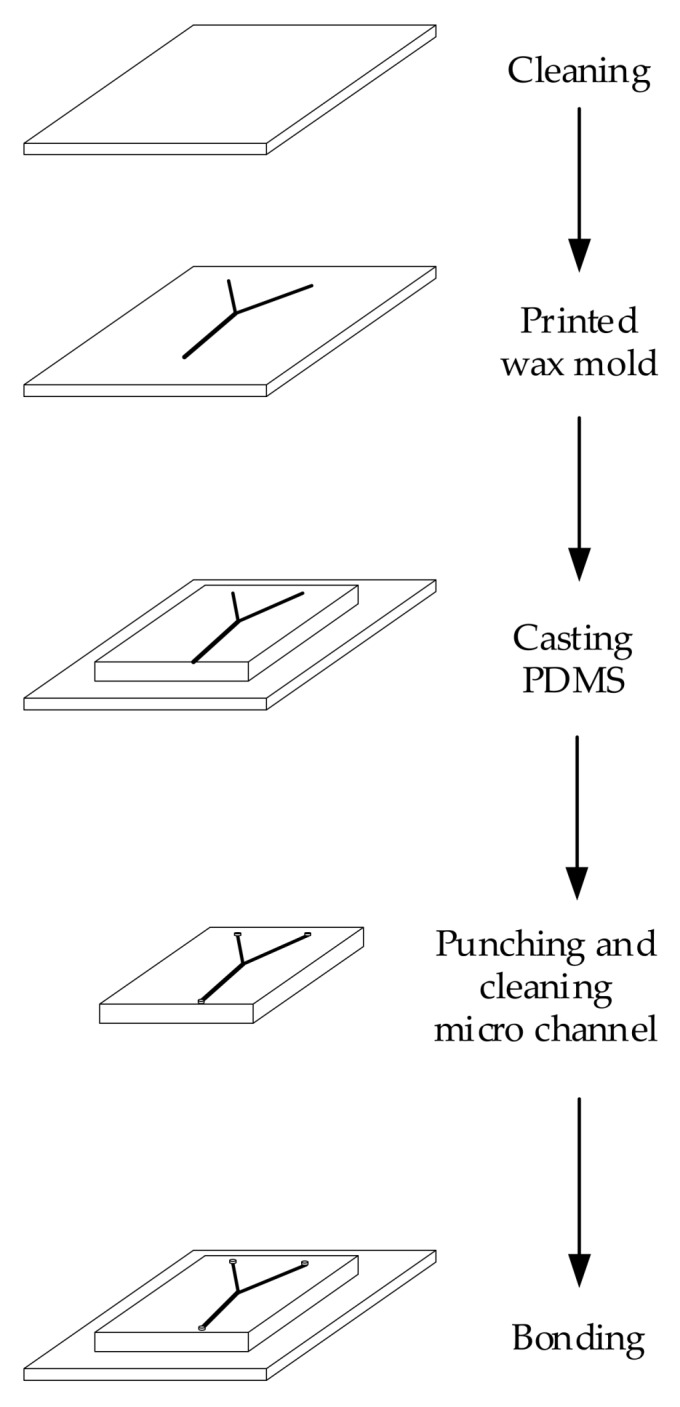
Process for the preparation of a polydimethylsiloxane (PDMS) microfluidic chip.

**Figure 4 sensors-20-01559-f004:**
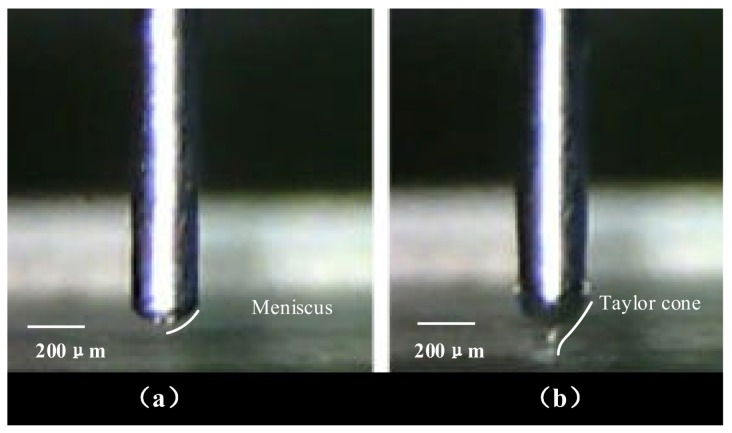
Photos of the droplet formation process on the nozzle: (**a**) a meniscus shape formed in the nozzle tip and (**b**) the Taylor cone formed under the action of the electric field force.

**Figure 5 sensors-20-01559-f005:**
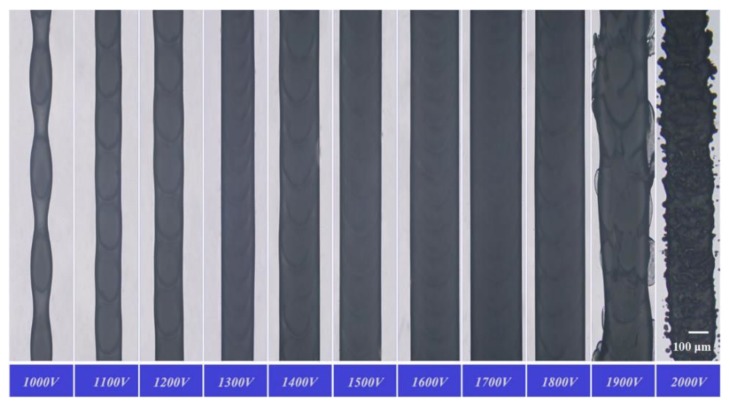
Photos of the printed wax lines with different working voltages.

**Figure 6 sensors-20-01559-f006:**
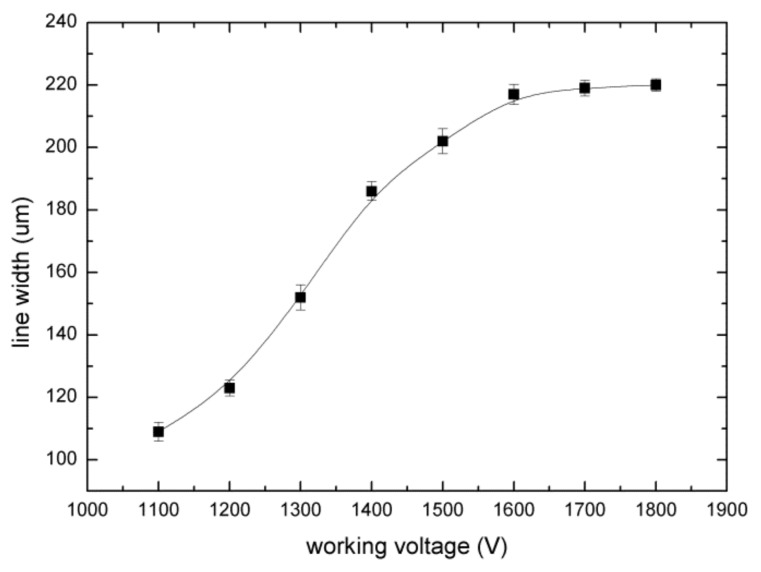
The relationship between the working voltage and the line width.

**Figure 7 sensors-20-01559-f007:**
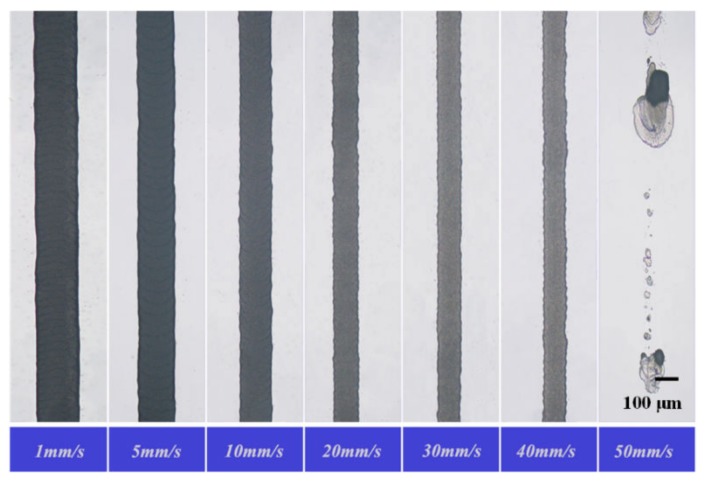
Photos of the printed wax lines with different movement speeds.

**Figure 8 sensors-20-01559-f008:**
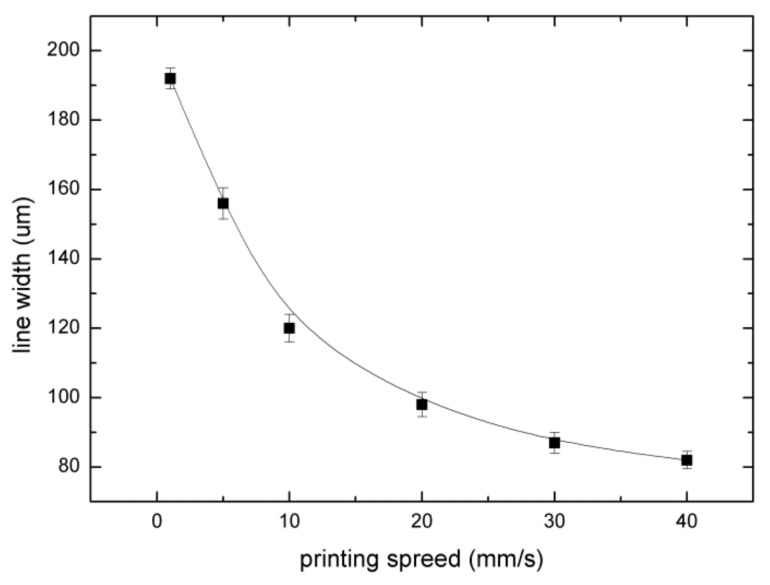
The relationship between the printing speed and the line width.

**Figure 9 sensors-20-01559-f009:**
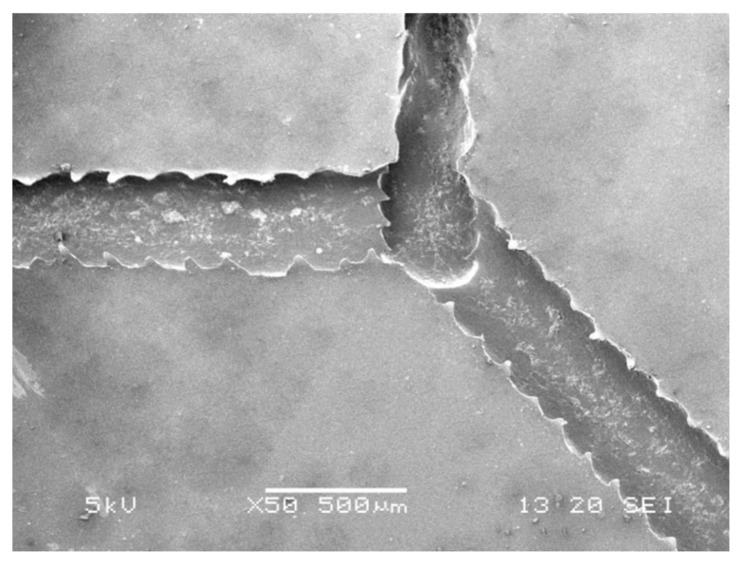
SEM photo of the intersection area of the PDMS microchannel.

**Figure 10 sensors-20-01559-f010:**
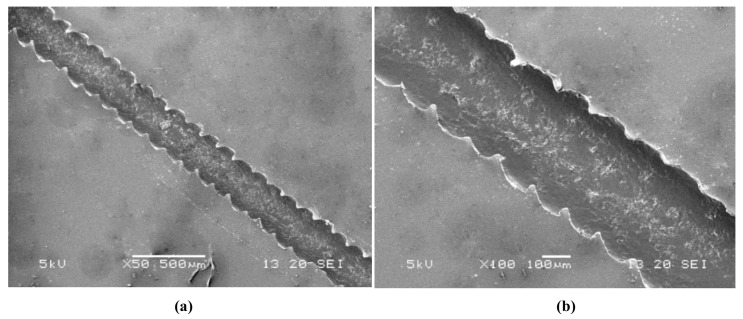
SEM photo of the straight area of the PDMS microchannel.

**Figure 11 sensors-20-01559-f011:**
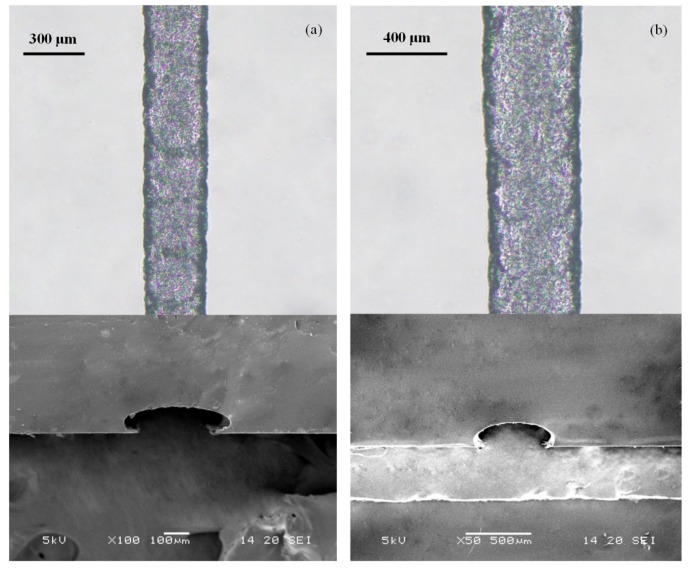
SEM photos of PDMS microchannels replicated using paraffin wax molds with different widths and heights: (**a**) the microchannel width was 270 μm and the height was 105 μm and (**b**) the microchannel width was 440 μm and the height was 180 μm.

**Figure 12 sensors-20-01559-f012:**
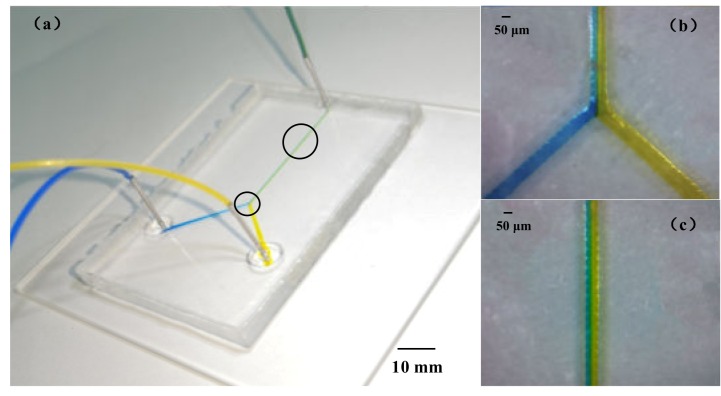
The Y-shaped microchannel: (**a**) mixing in the microchannel at a feeding speed of 10 μL min^−1^; (**b**) the mixed fluid at the junction; and (**c**) the mixed fluid near the outlet.

**Figure 13 sensors-20-01559-f013:**
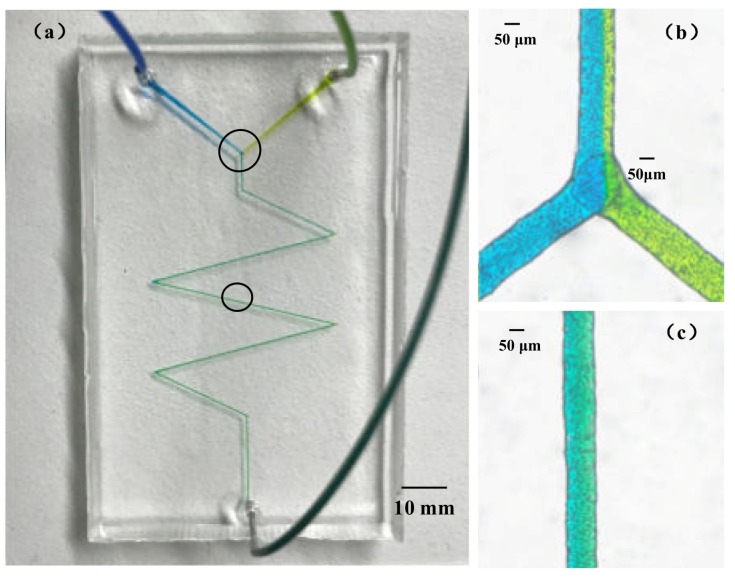
The zigzag microchannel: (**a**) mixing in the microchannel at a feeding speed of 10 μL min^−1^; (**b**) the mixed fluid at the junction; and (**c**) the mixed fluid near the outlet.

**Table 1 sensors-20-01559-t001:** The properties of the paraffin wax.

Molecular Weight	Density (g·mL^−1^)	Electrical Conductivity	Melting Point (°C)	Melt Viscosity (mm^2^·s^−1^)
650	0.8–0.92	insulation	45–55	10–20

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
