# Peer review of "Fabrication of Microfluidic Chips Based on an EHD-Assisted Direct Printing Method"

_sensors, 2020, doi:10.3390/s20061559_

Round 1

Reviewer 1 Report

The revised version appears improved but english errors are still present in the text, moreover some sentences still require a revision and are not clear.

Referring to the authors answer n.2 :

the sentence “Electrohydrodynamic (EHD) printing used the electrostatic force on the nozzle tip to form the Taylor cone, and the Pyro-EHD method is that when the area of the material is heated, it causes charge release to form the Taylor cone”  passing on the English (that should be revised) what the authors claim about Pyro-EHD is not correct. It seems that they should go through the details of this technology. I also do not understand why they removed the ref “ACS Applied materials and Interfaces 9(19), pp. 16488-16494” that was present in the previous version but leaved the description in the text (page 2 from line 57). I would suggest to leave this reference and add the new ones reported in the revised version as well.

Figure 4 is not readable, the taylor cone is not visible as claimed by the authors.

Page 7 from line 187 “However, as the rapid solidification of the molten paraffin wax when contacting the cold glass substrate, the edge of the microchannel appeared a zigzag pattern. This kind of structure can cause some imbalances of the flowing fluid and speed up the micromixing process.” Please add some reference and explain how it could be possible to control the flow in case of a zigzag profile.

The discussion (page 9) are recurring and not clear. “The moving speed has a obviously stronger influence than that of the working voltage as the adhesion force exists” the reason is not clear through the text

Authors should revise English all over the text very urgently and carefully and make some sentences more readable.

Author Response

      Thank you very much for your kind evaluation and comments on our paper. We have revised the manuscript according to your kind advices and detailed suggestions.

  1. The revised version appears improved but english errors are still present in the text, moreover some sentences still require a revision and are not clear.

Response: Thanks very much for the referee’s kind advice. We have revised the English all over the text with the help of MDPI.

  1. The sentence “Electrohydrodynamic (EHD) printing used the electrostatic force on the nozzle tip to form the Taylor cone, and the Pyro-EHD method is that when the area of the material is heated, it causes charge release to form the Taylor cone” passing on the English (that should be revised) what the authors claim about Pyro-EHD is not correct. It seems that they should go through the details of this technology. I also do not understand why they removed the ref “ACS Applied materials and Interfaces 9(19), pp. 16488-16494” that was present in the previous version but leaved the description in the text (page 2 from line 57). I would suggest to leave this reference and add the new ones reported in the revised version as well.

Response: Thanks for the referee’s kind advice. We have revised this part.

“the Pyro-EHD method is that when the area of the material is heated, it causes charge release to form the Taylor cone” was corrected as “in the Pyro-EHD method, the pyroelectric field of a lithium tantalate or lithium niobate crystal is activated by a wireless thermal source, and the printing material drop starts to deform into a Taylor cone under the action of the pyroelectric charges [22-24]. ” And we have added the reference “ACS Applied materials and Interfaces 9(19), pp. 16488-16494”[27] (see page 2 line 61) and added the refs “Opt Lett, 2012, 37(13): 2460-2462”[22], “ACS Omega, 2018, 3(12): 17707-17716”[23] and “Appl Phys Lett, 2015, 106(26): 261603”[24] (see page 2 line 55-57) reported in the revised version as well.

We also corrected other similar language corrections and sentences not easily understood, and marked them in the revised manuscript. Please read the revised manuscript in the revised model, and find them in the marked places. “Both these methods have the feature of high manufacturing accuracy, especially the Pyro-EHD method, which does not require electrodes, high-voltage circuit connections, or nozzles.”(see page 2 line 58-59)

  1. Figure 4 is not readable, the Taylor cone is not visible as claimed by the authors.

Response: Thanks for the referee’s kind advice. We have edited Figure 4 to make it readable. The Taylor cone has been marked out in the figure.

  1. Page 7 from line 187 “However, as the rapid solidification of the molten paraffin wax when contacting the cold glass substrate, the edge of the microchannel appeared a zigzag pattern. This kind of structure can cause some imbalances of the flowing fluid and speed up the micromixing process.” Please add some reference and explain how it could be possible to control the flow in case of a zigzag profile.

Response: Thanks for the referee’s kind advice. We have supplemented and modified this part and added relevant references. “The mixing of fluid usually depends on the generation of chaotic advection and/or turbulence, in which the fluid motion varies irregularly and thus causes quantities such as the pressure and velocity to vary randomly in both space and time. The simplest method of obtaining chaotic advection is to insert obstacles into the mixing channel. The zigzag structure could play the role of an obstacle to some extent and thus cause some imbalances of the flowing fluid and speed up the micromixing process [28, 29].”(see page 8 line 204-209)

  1. The discussion (page 9) are recurring and not clear. “The moving speed has a obviously stronger influence than that of the working voltage as the adhesion force exists” the reason is not clear through the text.

Response: Thanks for the referee’s kind advice. We have deleted and modified the duplicate content. The reason between the influences of moving speed and the working voltage on the wax line is indeed not clear.  We have deleted this sentence “The moving speed has an obviously stronger influence than that of the working voltage as the adhesion force exists”.

  1. Authors should revise English all over the text very urgently and carefully and make some sentences more readable.

Response: Thanks very much for the referee’s kind advice. We have revised the English all over the text and also searched for the help of MDPI.

     Thank you very much, and please see the attachment for the revised manuscript.

Reviewer 2 Report

The microfluids was usually prepared using the photoresist and PDMS, and the linewidth can be smaller than 50 micrometer very easily. However, the best results showed the linewidth larger than 100 micromiter. I hope the authors further improve the EHD method. Maybe the linewidth is hard to be reduced in a short time. The authors had better improve the structure showed in Fig. 11, because the mix is very poor from the experiment.

Author Response

   Thank you very much for your kind evaluation and comments on our paper. We have revised the manuscript according to your kind advices and detailed suggestions.  Please see the attachment for the revised manuscript.

  1. The microfluids was usually prepared using the photoresist and PDMS, and the linewidth can be smaller than 50 micrometer very easily. However, the best results showed the linewidth larger than 100 micromiter. I hope the authors further improve the EHD method. Maybe the linewidth is hard to be reduced in a short time. The authors had better improve the structure showed in Fig. 11, because the mix is very poor from the experiment.

Response: Thanks for the referee’s kind advice. This manuscript describes a printing process that is based on EHD Assisted Direct Printing Method. The resolution in our work ranged from ranging from 80 μm to 220 μm, and in the future work we will improve our method, such as heating the substrate or using pulse direct current voltage to improve the resolution. Also we have designed another kind of zigzag type micro mixer. The mixing channel was elongated and the direction of fluid was turned repeatedly to improve the mixing efficiency.

“In the Y-shaped and zigzag microchannel, the blue dye and yellow dye solutions showed a clear interface at the junction at a feed speed of 10 μL·min-1, as shown in figures 12 and 13. The area of the interface increased slowly as the fluid flowed; the color of the fluid near the outlet channel of the Y-shaped microchannel was a mixture of yellow and blue (Fig. 12c), and the color of the fluid was green in the outlet channel of the zigzag microchannel (Fig. 13c). It can be seen that the reasonable design of bending in the microchannel structure can make the fluid mix faster. This method is an fast and easy fabrication method for the preparation of wax molds used in the replication of PDMS microchannels.”(see page 9 line 232-239)

We also corrected other similar language corrections and sentences not easily understood, and marked them in the revised manuscript. Please read the revised manuscript in the revised model, and find them in the marked places.

Reviewer 3 Report

In this manuscript, the authors proposed an EHD assisted direct printing method for fabricating sacrifice paraffin wax mold. The influences of operating voltage and moving speed of the work table on the width of the paraffin wax mold were studied. They also fabricated a paraffin wax mold for a microfluidic mixer which realized the passive mixing of the blue and yellow dyes. This is an overall and systematic work, and the manuscript is clearly presented. More importantly, the EHD assisted direct printing technology presented in this manuscript provides a potential way for precise microchannel fabrication. A few minor comments are listed below:

  1. The supplementary Figure S1 and Figure S2 are strongly suggested to be listed in the manuscript. As the experimental results in the two figures show the quantitative effects of the printing parameters on the wax line, it is helpful for the readers to obtain the information directly from the manuscript.
  2. The two images of the microchannel cross-sections in Figure 9 (a) and (b) should be shown in the same direction and same type for clear comparison.
  3. The authors studied the influences of the printing parameters on the width of the wax line. However, the height of the wax line was not investigated. As channel height is very important for a microfluidic device, the authors should at least discuss the channel height influenced by the printing process.
  4. I’m curious about the bonding strength of the PDMS device. What about the bonding performance of the device by using the bonding parameters in the manuscript?
  5. There are some spelling and grammar mistakes in the manuscript, I suggest the authors find a native English speaker to proofread the whole manuscript.

Author Response

Thank you very much for your kind evaluation and comments on our paper. We have revised the manuscript according to your kind advices and detailed suggestions.  Please see the attachment for the revised manuscript.

  1. The supplementary Figure S1 and Figure S2 are strongly suggested to be listed in the manuscript. As the experimental results in the two figures show the quantitative effects of the printing parameters on the wax line, it is helpful for the readers to obtain the information directly from the manuscript.

Response: Thanks very much for the referee’s kind advice. We have added the supplementary Figure S1 and Figure S2 in the manuscript, and rearrange the numbers of the figures.

  1. The two images of the microchannel cross-sections in Figure 9 (a) and (b) should be shown in the same direction and same type for clear comparison.

Response: Thanks very much for the referee’s kind advice. We have revised Figure 9, and rename it to figure 11.

  1. The authors studied the influences of the printing parameters on the width of the wax line. However, the height of the wax line was not investigated. As channel height is very important for a microfluidic device, the authors should at least discuss the channel height influenced by the printing process.

Response: Thanks very much for the referee’s kind advice. We discussed the channel height influenced by the printing process and added it to the conclusions.

“Paraffin wax lines with width ranging from 80 μm to 220 μm and height ranging from 35 μm to 90 μm can be prepared using this method. In this range, the width and height of the paraffin wax line increase with increasing working voltage. ”(see page 10 line 255-257)

We also corrected other similar language corrections and sentences not easily understood, and marked them in the revised manuscript. Please read the revised manuscript in the revised model, and find them in the marked places.

  1. I’m curious about the bonding strength of the PDMS device. What about the bonding performance of the device by using the bonding parameters in the manuscript?

Response: Thanks very much for the referee’s kind advice. The PDMS Microfluidic chips mentioned in this study were plasma treated before being bonded with the glass chip with the help of Suzhou Wenhao Microfluidic Technology Co., Ltd., (http://www.whchip.com) and this kind of bonding strength can range from 40 psi to 70 psi  as shown in the reference “Journal of microelectromechanical systems, 14(3), 590-597”.

  1. There are some spelling and grammar mistakes in the manuscript, I suggest the authors find a native English speaker to proofread the whole manuscript.

Response: Thanks very much for the referee’s kind advice. We have revised the English all over the text and also searched for the help of MDPI.

Round 2

Reviewer 2 Report

Although many problems still exists, the EHD method is interesting.

This manuscript is a resubmission of an earlier submission. The following is a list of the peer review reports and author responses from that submission.

Round 1

Reviewer 1 Report

This manuscript describes a printing process that is based on electric field-assisted-extrusion printing. This is not the so-called electrohydrodynamic printing. The electric field assists in shaping the meniscus to touch the substrate. The study was not conducted in a rigorous way and the printing resolution was comparable to that obtained by extrusion-based printing. The demonstration of printing of a mold to make microfluidic devices is over simplified. The reviewer could not see much scientific advance in this manuscript. 

Reviewer 2 Report

In this paper Direct-Writing method based on electrohydrodynamic (EHD) printing was proposed for the fabrication of microfluidic chip. In its current form the article is very difficult to read, it seems to be written quickly without a revision between the authors. The English is very poor and sometimes it is hard to follow the description.  In general the topic proposed could be of interest for researchers involved in the field of direct writing and microfluidic but, in my opinion it has to be totally revised, adding some additional information and details.

The introduction should be revised, probably a revision by a English speaker could help the authors. Moreover, they proposed a method based on the EHD printing for the fabrication of microfluidic chip but in the same introduction they made some references to other methods based on EHD (see page 2 line 59). Probably I will expect some line of comments describing the difference of the method proposed with the other ones. I would also suggest to extract reference 35 form the others, this work is, in fact related to a different way of the activation of the EHD effect and in my opinion the reader would benefit in a new sentence focused on this approach (see also ACS Omega 3(12), pp. 17707-17716, 2018 and Applied Physics Letters 106(26),261603, 2015). At the end of the introduction, I will expect to read the major advantages related to this process. page 2 line 76, please use the symbol for degrees page 3 lines 90-92 are repetitive of lines 85-87, it was a copy and paste made two times. page 3 line 97, the authors refer to “double heating integrated device”, while they choose a double heating process? How it is possible to control it? Please add some more details There is a lot of confusion in the text, sometimes the authors use the term jet/ printing and some other the term spray. They are based on different physical and experimental approaches, for this reason I suggest to decide if they are using a direct printing method based on jet or spray? Pleas add some physical details on the jetting process page 4 line 108: paraffin or wax? English is very poor all along the text just an example page 4 lines 112-115: the period is divided in two parts but the division doesn’t work Please add a more detailed description of the experimental parameters used in the fabrication process. page 4 line 121, all the acronyms must be defined once they appear for the first time. I read:…the liquid silicone elastomers A and B….and some sentences after I read PDMS, please revise the entire description

  12.page 6 line 165: spraying process, probably it should be revised in       printing process.

The authors stated the they printed 3D fibers, did they have some characterization of their 3D profile? Could they furnish a measure of height of the so fabricated stamps? Did authors make some experiments with solution more viscous? What is the limit of processing polymeric solutions? Authors should revise English all over the text very urgently and make some sentences more readable.
